# Sodium Alginate-Based Green Packaging Films Functionalized by Guava Leaf Extracts and Their Bioactivities

**DOI:** 10.3390/ma12182923

**Published:** 2019-09-10

**Authors:** You Luo, Haiqing Liu, Shanzhong Yang, Jiarui Zeng, Zhenqiang Wu

**Affiliations:** 1School of Biology and Biological Engineering, South China University of Technology, Guangzhou 510006, China; 2Department of Biological Engineering, Pan Asia (Jiangmen) Institute of Biological Engineering and Health, Jiangmen 529080, China

**Keywords:** bioactive film, sodium alginate, guava leaf extract, polyphenol, antioxidant, antibacterial

## Abstract

The aim of this work was to develop green and bioactive films with sodium alginate incorporating guava leaf extracts. Seven formulations were performed with a different sodium alginate: Guava leaf water extract (WE)/ethanolic extract (EE) proportions (100:0, 90:10, 85:15, 80:20), and glycerol were used as a plasticizer. The HPLC-PDA analysis showed the main phenolic compounds in WE were gallic acid, ellagic acid, quercetin-3-O-β-D-xylopyranoside, avicularin and quercetin. The main polyphenols in EE were rutin, isoquercitrin, quercetin-3-O-β-D-xylopyranoside, avicularin, quercitrin, quercetin and kaempferol. Guava leaf extracts could greatly enhance the antioxidant activity, antibacterial activity, tensile strength and water solubility of the sodium alginate film as well as the water barrier property, while inducing a decrease in the moisture content and elongation at the break. The FTIR and SEM analyses indicated that intermolecular hydrogen bonding between the guava leaf extract and sodium alginate resulted in a more compact structure in the composite films. These results indicated that sodium alginate-guava leaf extract films might be developed into antiradical and antimicrobial food packaging materials.

## 1. Introduction

Worldwide, 8.3 billion tons of plastics have been produced, of which 6.3 billion tons have become plastic waste, causing serious environmental pollution [1]. Now the whole world is declaring war on plastic and seeking alternatives to plastics. Biodegradable and edible food packaging materials have become a research hotspot due to the serious white pollution and health concerns [2,3]. Biopolymers, such as polysaccharides, protein and lipid have been developed into food packaging materials [4,5,6]. Among the varieties of biopolymers, sodium alginate, a natural polysaccharide, is a good candidate for a biopolymer film or coating due to the excellent colloidal properties such as its film-forming property, stability, gelation and biocompatibility [7,8]. Furthermore, the active packaging film is designed to extend the shelf life of a product and maintain nutrition, freshness and safety [9]. The main factors of food spoilage are oxidation and microorganisms. Hence, active packaging should include antioxidants and bacteriostats to prevent unfavorable factors. 

Plant polyphenols and extracts are natural antioxidants and bacteriostats. The antioxidant mechanism of polyphenols mainly depends on their abilities for scavenging reactive oxygen species and chelate metal ions [10]. The bacteriostatic effects of plant-derived polyphenols depend upon the hydroxyphenyl structure [11]. Therefore, polyphenols or plant extracts, such as tea polyphenols [12], green apple skin extracts [9] and thyme extract [13], can greatly promote the antioxidant, antibacterial and physical properties of film. From the point of view of resource recycling and environmental protection, it seems more profitable to utilize agricultural by-products, such as peel extracts and plant leaf extracts, as active components of the films. Guava (*Psidium guajava Linn.*) is a common fruit plant available in many regions in tropical and subtropical climates. Guava leaf has been used as a herbal tea in China, Japan, India, et al. for many years because of its extensive bioactivities and pharmacological effects [14,15]. Previous studies have verified that the extracts from guava leaves exhibited antioxidant and antibacterial activities [16,17,18].

Yang et al. (2018) improved the chemical and mechanical properties of alginate-based films by adding polyvinyl alcohol and SiO_2_ [19]. Costa et al. (2018) verified that the tensile strength of sodium alginate-based films crosslinked with CaCl_2_ increased, and the moisture content and water solubility decreased [20]. Salama et al. revealed that chitosan biguanidine incorporated in sodium alginate markedly elevated the antibacterial ability of the films [21]. However, there are a few reports about the physical properties and bioactivities of sodium alginate/phenolic-rich plant extracts composite films. The purpose of this work is to prepare antioxidant and antibacterial sodium alginate-based films by adding different amounts of guava leaf water extract (WE)/ethanolic extract (EE). Firstly, the sodium alginate/guava leaf extracts composite films were prepared. Then, the physical properties, antioxidant and antibacterial activities of those composite films were assessed and compared with plain sodium alginate film.

## 2. Materials and Methods

### 2.1. Materials

Firstly, 1,1-Diphenyl-2-picrylhydrazyl (DPPH) and sodium alginate (product number: S817374; AR, purity ≥90%; viscosity (1%, 20 °C), 260 mPa·s) were purchased from Macklin Biochemical Co., Ltd. (Shanghai, China). Glycerol and Absolute ethanol (AR) were obtained from Tianjin Zhiyuan Chemical Reagent Co., Ltd. (Tianjin, China). Gallic acid, chlorogenic acid, *p*-hydroxybenzoic acid, caffeic acid, rutin, isoquercitrin, quercetin-3-O-α-L-arabinoside, ellagic acid, avicularin, quercetin, quercitrin, kaempferol were purchased from Aladdin Industrial Corporation (Shanghai, China). 

*Staphylococcus aureus* (ATCC29215) and *Escherichia coli* (ATCC25922) were provided by the Microbiology Laboratory of South China University of Technology (Guangzhou, China).

Guava leaves were collected in October, 2018 and supplied by Jiangmen Nanyue Guava farmer cooperatives (Jiangmen, Guangdong, China). The guava leaves were naturally dried and stored in the dark (25 ± 2 °C). Dried guava leaves were grounded into powders and passed through an 80 mesh sieve size. 

### 2.2. Obtaining Guava Leaf Extracts

The guava leaf powder (10 g) was immersed in 100 mL of ethanol/water (70%, v/v) and distilled water, respectively. The extraction was conducted with an AS20500ATH ultrasonic cleaner bath (400 W, 40 KHz, Tianjin Automatic Science Instrument Co., Ltd., Tianjin, China) at room temperature for 30 min. The mixture was centrifuged for 10 min and then the supernatant was obtained. The ethanolic extract (EE) and water extract (WE) as functional ingredients were added to the films.

### 2.3. Films Preparation

The films were prepared by casting with sodium alginate (SA), guava leaf extracts and glycerol (as plasticizer). The SA was dissolved in distilled water (1:100 g/mL) at 90 °C under constant stirring for 1 h using a B11-2 constant temperature magnetic stirrer (450 r/min, Shanghai Sile Instrument Co., Ltd., Shanghai, China). Glycerol (Gly, 0.5 g) was added into the SA solution. The different film formulations were illustrated in Table 1. The obtained films were entitled SAF (sodium alginate film), 10% WEF (sodium alginate incorporating 10% water extract film), 15% WEF (sodium alginate incorporating 15% water extract film), 20% WEF (sodium alginate incorporating 20% water extract film), 10% EEF (sodium alginate incorporating 10% ethanol extract film), 15% EEF (sodium alginate incorporating 15% ethanol extract film), 20% EEF (sodium alginate incorporating 20% ethanol extract film), respectively. 

The film-forming solutions (50 mL) were poured into 10 × 10 cm square Petri dishes and dried in a 50 °C oven. The formed films were kept at 20 °C and 50% relative humidity (RH) in a constant temperature and humidity incubator for 3 days before further tests.

### 2.4. Characterization

#### 2.4.1. Chemical Composition of Extract

The total polyphenol content of the extract was measured by the Folin-Ciocalteu method [22]. The reaction system consisted of the Folin-Ciocalteu reagent, 20% Na_2_CO_3_ and the sample. The absorbance was read at 765 nm by a Multimode Plate Reader (PerkinElmer, Singapore). The standard curve of polyphenol was plotted with gallic acid as the standard. The total phenolics content was expressed in milligrams gallic acid equivalent (GAE) per milliliter of the extract.

The extract of guava leaf and phenolic standards (gallic acid, chlorogenic acid, *p*-hydroxybenzoic acid, caffeic acid, rutin, isoquercitrin, quercetin-3-O-α-L-arabinoside, ellagic acid, avicularin, quercetin, quercitrin, kaempferol) were performed on SunFire C18 column (250 × 4.6 mm, 5 μm, Waters, USA) and detected by high performance liquid chromatography (HPLC) with photo-diode array (PDA) (Waters, USA). The mobile phase A and phase B were 0.1% formic acid solution and acetonitrile, respectively. The elution condition: 0–5 min, 15% B; 5–10 min, 15–20% B; 10–20 min, 20–25% B; 20–30 min, 25–35% B; 30–40 min, 35–50% B; 40–50 min, 80% B; 50–55 min, 15% B. The flow rate was 0.8 mL/min, the injection volume was 10 μL, the column temperature was 30 °C, and the detection wavelength was 200–600 nm.

#### 2.4.2. Fourier Transform Infrared (FT-IR) Analysis

The film was mixed with KBr and submitted to the FT-IR analysis. The FT-IR spectra were determined by a Nexus Por Euro Fourier transformation infrared spectrometer (Nicolet, Madison, WI, USA) in the range of 4000–400 cm^−1^. 

#### 2.4.3. Scanning Electron Microscopy (SEM)

The film samples were cryo fractured in liquid nitrogen and sprayed with gold. The microstructure of the film sample was observed by SEM (Zeiss, Oberkochen, Germany) at a voltage of 5 kV. 

#### 2.4.4. Moisture Content and Water Solubility 

The moisture content (MC) was determined by the weight loss of the film after drying to a constant weight at 105 °C. The moisture content was presented as g of water per 100 g of the sample.

The sample film (1.5 cm × 1.5 cm) was weighed accurately (W_0_) and soaked in 10 mL of deionized water. All samples were stirred continuously for 1 h at room temperature (25 ± 1 °C). After centrifugation, the residue was dried to a constant weight (W_1_) at 105 °C [23].

(1)$$S\%\;=\;\frac{\left(W0-W1\right)}{W0}\times 100\%$$

#### 2.4.5. Film Thickness

The thickness of the films was measured with a digital micrometer (Sanliang Measuring Tools Co., Ltd. Dongguan, China) with an accuracy of 0.001 mm. The average thickness was calculated by 6 random points on the film.

#### 2.4.6. Color Measurement

The color parameters including L (lightness), a (red/green) and b (yellow/blue) of the films were measured with a Chroma meter CR-400 (KONICA MINOLTA OPTICS, INC. Tokyo, Japan). The colorimeter was calibrated using a standard white plate (L* = 95.19, a* = −0.57, b* = 3.53). The color parameters were determined by three random points on each film.

#### 2.4.7. Water Vapor Permeability (WVP)

The WVP value of the film was measured according to the ASTM (American Society for Testing and Materials) method [24]. Each film covered a permeation cell with an opening of 1.5386 cm^2^ at 20 °C in a dryer. In order to maintain a 75% RH gradient across the film, anhydrous CaCl_2_ was placed in the cell, and saturated NaCl solution was placed in the dryer. The water vapor transport depended on the weight gain of the cell. After reaching the steady state condition, the cells were weighed every 2.5 h for 25 h. The change of cell weight was accurately recorded to 0.0001 g and plotted as a time function. The slope of each line was calculated by linear regression (r^2^ >0.99), and the water vapor transmission rate (WVTR) was calculated as the slope (g/s) divided by the film area (cm^2^). WVP (g·Pa^−1^·s^−1^·cm^−1^) = [WVTR/S(R_1_ − R_2_)]d, where S is the saturation vapor pressure of water (Pa) at the test temperature (20 °C), R_1_ is the RH in the dryer, R_2_ is the RH in the permeation cell, and d is the film thickness (cm). The driving force [S(R_1_ − R_2_)] was 1753.55 Pa. Each sample was repeated three times.

#### 2.4.8. Mechanical Properties

The stress and strain analysis of the films was carried out by a DMA Q800 (TA Instruments, Newcastle, DE, USA) with a tension clamp. Three sample strips (8 × 30 mm) of each formulation were tested using a ramp force constant rate of 0.4 N/min to a static force of 18 N, and the samples were ruptured. The force (N) and deformation (mm) were recorded during extension. The tensile strength (MPa) and elongation at break (%) were obtained from the stress-strain curves.

#### 2.4.9. Antiradical Activity

The antiradical ability of the films was assessed by DPPH assay following Shimada’s method with slight modification [25]. On the one hand, the different weights of the films (10, 15, 20, 25 and 30 mg, respectively) were put into 3 mL of 0.1 mM DPPH solution. The reaction was carried out at room temperature in darkness for 60 min. The absorbance of the reaction solution at 517 nm was measured by a microplate reader (PerkinElmer, Singapore). For the time-dependent antioxidant assay, 30 mg of each film was put into 3 mL of 0.1 mM DPPH solution. The absorbance was read at 10, 20, 30, 40, 50 and 60 min, respectively. The scavenging DPPH radical ability was calculated as follows:(2)$$\mathrm{DPPH}\;\mathrm{radical}\;\mathrm{scavenging}\;\mathrm{rate}\;\left(\%\right)=\frac{\mathrm{Ab}-\mathrm{As}}{\mathrm{Ab}}\times 100$$
where Ab is the absorbance at 517 nm of the methanol solution of DPPH, as is the absorbance value at 517 nm of the sample.

#### 2.4.10. Antimicrobial Properties

The antimicrobial experiment was conducted according to the method reported by Otoni et al. with changes [26]. *Escherichia coli* and *Staphylococcus aureus* were cultured in Luria-Bertani (LB) culture for 8 h. The number of microorganisims was calculated by a hemocytometer. Further, 200 μL of inoculant containing 10^6^ CFU/mL of the microorganism was uniformly coated on a nutrient agar medium. Then, the 8 mm diameter films were placed on the respective culture medium and kept at 37 °C for one day. The diameter (millimeter) of the inhibition zones was measured with a caliper. Each sample was repeated three times.

### 2.5. Statistical Analysis

The difference between the factors and levels was analyzed by ANOVA using SPSS 17.0. The Duncan test was used to compare the differences among the groups (*p* < 0.05). All data were list as the mean ± standard deviation. The different comparisons among the groups were marked with different letters [12]. The average of each group was arranged from large to small. The group with the maximum average was marked with the letter “a”, and then compared with the other averages. If the difference is not significant, the group is marked the same letter. If the difference is significant, the group is marked with another letter. 

## 3. Results and Discussion

### 3.1. Composition Difference between Water Extract and Ethanolic Extract of Guava Leaf

The polyphenol content of each 1 mL water extract and ethanolic extract was 1.84 mg and 2.94 mg gallic acid equivalent, respectively. Polyphenols in guava leaf extracts were further analyzed by HPLC combined with PDA. The peaks were identified by the retention time, UV-Vis and mass spectra together with our previous work [17]. As shown in Figure 1, several principal components were detected at the wavelength of 350 nm. The phenolic compounds in the guava leaf water extract were gallic acid (0.0029 μg/mL), ellagic acid (0.0373 μg/mL), quercetin-3-O-β-D-xylopyranoside (0.0057 μg/mL), avicularin (0.0881 μg/mL) and quercetin (0.0001 μg/mL). The main polyphenols in ethanolic extract were rutin (0.0015 μg/mL), isoquercitrin (0.0291 μg/mL), quercetin-3-O-β-D-xylopyranoside (0.1290 μg/mL), avicularin (0.1225 μg/mL), quercitrin (0.0818 μg/mL), quercetin (0.0332 μg/mL) and kaempferol (0.0050 μg/mL). The results indicated that the types of phenolic compounds in these two extracts were different and the content of polyphenols in EE was much higher than WE.

### 3.2. FTIR Spectra of Films

The location and intensity of the characteristic absorption peaks of SAF, WEF and EEF are shown in Figure 2. The FTIR spectra of SAF and WEF showed very similar characteristic peaks. For SAF, a broad absorption band at 3250 cm^−1^ was attributed to the O-H stretching vibration which was affected by the inter-molecular or intra-molecular hydrogen bonds [27]. The peaks at 2940 cm^−1^, 1600 cm^−1^ and 1400 cm^−1^ were assigned to C–H stretching [27], C=O vibration [28] and C–H stretching [29] respectively. The absorption ranged from 1260 to 1000 cm^−1^ and was due to the stretching vibrations of the C–O–C and C–OH groups. The sodium alginate film had a strong absorption peak at 1030 cm^−1^ due to the presence of the glycosidic bond (C–O–C) [28]. WEF exhibited characteristic bands at 3280 cm^−1^, 2940 cm^−1^, 1600 cm^−1^, 1400 cm^−1^ and 1020 cm^−1^. An obvious decrease and a slight shift of the band can be observed from 3280 cm^−1^ to 3250 cm^−1^. This phenomenon also occurred in EEF, the absorption band of O-H significantly decreased and shifted from 3260 cm^−1^ to 3250 cm^−1^. In addition, those absorption peaks located at 2920 cm^−1^, 1590 cm^−1^ and 1020 cm^−1^ had a slight shift in EEF from that of SAF, which was indicative of the interactions between the sodium alginate and extracts from guava leaves. Previous studies indicated that the changes of the peak position and intensity were induced by hydrogen bonds between polyphenols and sodium alginate [30,31]. Stronger intermolecular interactions could improve the physical properties of the films, such as the water barrier property and tensile strength.

### 3.3. Microstructure of Films

The microstructures of the films were magnified 50 times and 8000 times by SEM, respectively, as shown in Figure 3. From the magnified images of 50 times, it can be seen that the surface of all the grinded film samples were relatively smooth and the cross section was compact. From the magnified images of 8000 times, there were bulges and several cracks in the plain sodium alginate film, which could exert some adverse effects on the film properties such as, moisture resistance and the mechanical property. This result confirmed the fact that SAF possessed poor moisture resistance and tensile strength. After incorporation of the guava leaf extract, the composite films, especially EEF, appeared more compact and continuous owing to the formation of the hydrogen bond between the guava leaf extract and sodium alginate. Dou et al. observed a continuous surface without holes or friable areas, and the denser structure of gelatin-sodium alginate films with tea polyphenol [32]. The low WVP value and the better mechanical property were attributed to the more compact structure.

### 3.4. Film Appearance

Color is a significant attribute of the product, which affects the product appearance and uses. As illustrated in Figure 4, sodium alginate-WE films became yellow with the addition of WE and the color deepened with the higher content of WE. The sodium alginate-EE films became brown with the incorporation of EE and the color also deepened with the increasing EE content. The color parameters of the different films are listed in Table 2. The films incorporating WE and EE had lower L values than the plain sodium alginate film, which were relatively dark (*p* < 0.05). The higher concentration of the guava leaf extract was, the lower L value obtained. Nevertheless, the b values evidently increased (*p* < 0.05). This indicated the color of the film tended to yellowness. Except for 20% EEF, the A values of the other films were positive, indicating the tendency toward redness. The total color differences (ΔE) of all the composite films were markedly higher than the plain sodium alginate film (*p* < 0.05). Dou et al. found that tea polyphenols reduced the L values and increased the A and B values of the composite films as compared with the gelatin-sodium alginate film [32]. Considering the opacity of the composite films, they can be applicable for foods susceptible to light exposure.

### 3.5. Moisture Content, Water Solubility, Thickness and Water Vapor Permeability

The moisture content, water solubility, thickness and water vapor permeability of the films are presented in Table 3.

The moisture content of the composite films remarkably reduced compared with the control sodium alginate film, ranging from 66.92% to 17.24% (*p* < 0.05). The plain sodium alginate film presented a much higher moisture content due to the abundant hydrophilic groups (–OH and –COOH) in the sodium alginate molecules. When WE or EE was added into sodium alginate, the hydroxyl groups in the extracts could form intermolecular hydrogen bonds with the hydrophilic groups in sodium alginate. Therefore, the interaction between the water and composite matrix was reduced. A similar variation tendency of the moisture content was also reported by Wu et al. when pomelo peel flours films incorporated tea polyphenol [12].

Water solubility is a reflex of the water resistance of the films and coatings. The plain sodium alginate film exhibited the lowest water solubility (*p* < 0.05). The water solubility of the films became better with the increase of the WE/EE content. The improved water solubility was attributed to the hydrophilic groups of polyphenols in WE and EE that can easily bind with water. A similar trend occurred in the chitosan films with the addition of apple polyphenols [31].

The composite films were much thinner than the plain sodium alginate film (*p* < 0.05). With the increasing of WE concentration, the thickness of the sodium alginate-WE films decreased. In contrast, the thickness of the sodium alginate-EE films increased with the increasing of the EE concentration because of the more amount of solid contents.

WVP reflects the ability of the film to prevent moisture transfer [33]. WVP is influenced by many factors such as thickness, components, humidity, water activity and so on [34]. The composite films exhibited much lower WVP values than the plain sodium alginate film (*p* < 0.05). Further, 10% EEF reached the minimum WVP value of 0.55 × 10^−11^g·Pa^−1^·s^−1^·cm^−1^, which significantly decreased by 69.27% compared with SAF. The results revealed the incorporation of the guava leaf extract could greatly improve the water-resistant property of the sodium alginate film. Previous studies reported that the dense networks formed through intermolecular interactions between polyphenols and biopolymers contributed to the lower WVP values of the composite films [12,33]. The WVP values of sodium alginate-WE films significantly decreased when the WE content increased from 10 to 20% (*p* < 0.05). The sodium alginate-EE films gradually increased the WVP values with more addition of EE (*p* < 0.05). It is likely that the strong interfacial interaction between phenolics and sodium alginate could improve moisture resistance. However, the high content of phenolics might accumulate in the polymer matrix to produce voids, leading to higher WVP values [12]. 

### 3.6. Mechanical Properties of Films

The mechanical strength and flexibility of the film are usually reflected by the tensile strength and elongation at the break [35]. The mechanical properties of the plain sodium alginate films and the composite films are summarized in Table 4. The addition of the guava leaf extract caused a significant increase in the tensile strength and a significant decrease in the elongation at the break (*p* < 0.05) compared to the plain sodium alginate film. Overall, 10% WEF had preferable mechanical properties, tensile strength and elongation which were 10.86 MPa and 22.47%, respectively. The value of the tensile strength and elongation declined with increasing the WE proportion, while the mechanical properties of EEF were less affected by the extract. As the proportion of EE increased, the tensile strength of EEF increased first and then decreased, while elongation decreased. Dou et al.’s research also showed that the gelatin-sodium alginate film with higher tea polyphenols were stronger but more brittle [32].

Owing to the interfacial hydrogen interactions between sodium alginate and the guava leaf extract, a more compact structure of the composite film was obtained. Thus, the tensile strength of the composite film was enhanced. However, the tensile strength of the films decreased at a higher proportion of the extracts. The excessive proportion of the extracts may result in uneven dispersion in the mixture. 

### 3.7. Antioxidant Properties of Films

The composite films displayed strong DPPH free radical scavenging ability. Except for the SA film, the composite films presented dose-independent DPPH radical scavenging activities in Figure 5A. The DPPH radical scavenging rate of the different composite films significantly increased with the increasing amount of the films from 10 mg to 30 mg (*p* < 0.05). Meanwhile, when the proportion of the guava leaf extract increased from 10% to 20%, the DPPH radical scavenging abilities of the films were also improved to a certain extent. Conversely, the plain sodium alginate film showed a weak DPPH radical scavenging ability. As shown in Figure 5B, all composite films exhibited time-dependent DPPH radical scavenging capabilities. The clearance rate markedly increased within 30 min, then slowed down, and reached the maximum at 50 min. The order of the DPPH free radical scavenging ability of these films was as follow: EEF > WEF > SAF. The differences of antioxidant activity among the films were ascribed to the kinds and contents of polyphenols. Polyphenols in ethanolic extract are more abundant than those in the water extract. Similar enhancements in scavenging radical assays were also attained when apple polyphenol, maqui berry extract and black soybean seed coat extract were added into the chitosan film [31,36,37]. Therefore, the guava leaf extract can greatly improve the anti-oxidative capacity of the packaging materials to prevent food from oxidative deterioration.

### 3.8. Antibacterial Properties of Films

The antibacterial capacity of the different films against *E. coli* and *S. aureus* is presented in Figure 6 and Table 5. The plain sodium alginate films exhibited a certain antimicrobial ability against *E. coli* and *S. aureus*. With increasing the proportion of the guava leaf extract, the diameter of the inhibition zones gradually increased, especially EEF (*p* < 0.05). Further, 10% EE could greatly improve the inhibitory effect of the SA film on *E. coli* and *S. aureus*. 15% WE markedly improved the inhibitory effect of the SA film on *E. coli* (*p* < 0.05). Furthermore, 10% WE significantly enhanced the inhibitory effect of the SA film on *S. aureus* (*p* < 0.05). The SA/EE films showed better antibacterial properties than the SA/WE films. On the one hand, the polyphenol content of the ethanolic extract from guava leaves was 1.6 times higher than the water extract. On the other hand, the different kinds of polyphenol exert different influences on bacteria, and the polyphenol mixture is generally much more active than either polyphenol alone. It was reported that the combinations of rutin and quercetin, quercetin and quercitrin, kaempherol and rutin showed better antibacterial activity than either flavonoid alone [38]. Wu et al. observed that the addition of tea polyphenols significantly promoted the antimicrobial property of the composite film compared with the control [12].

## 4. Conclusions

Bioactive films were successfully developed by adding the water extract/ethanolic extract from the guava leaves into sodium alginate. The guava leaf extracts significantly enhanced the antioxidant and antibacterial abilities as well as the tensile strength and water-resistant properties of the composite films when compared to the plain sodium alginate film. The water resistance, tensile strength, antioxidant activity and antibacterial activity decreased in the order of EEF > WEF > SAF. The results provide a strategy to utilize agricultural by-products which provide functional ingredients. The results can further provide an insight to active packaging for food.

## Figures and Tables

**Figure 1 materials-12-02923-f001:**
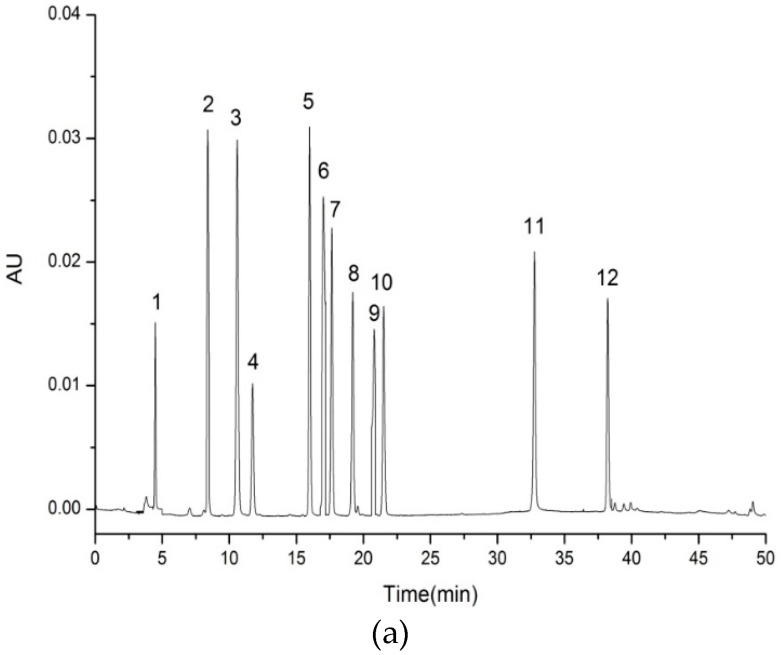
HPLC spectra of polyphenol standards (**a**) and polyphenols in guava leaf water extract/ethanolic extract (**b**); 1. Gallic acid, 2. Chlorogenic acid, 3. *P*-hydroxybenzoic acid, 4. Caffeic acid, 5. Rutin, 6. Ellagic acid, 7. Isoquercitrin, 8. quercetin-3-O-β-D-xylopyranoside, 9. Avicularin, 10. Quercitrin, 11. Quercetin, 12. Kaempferol.

**Figure 2 materials-12-02923-f002:**
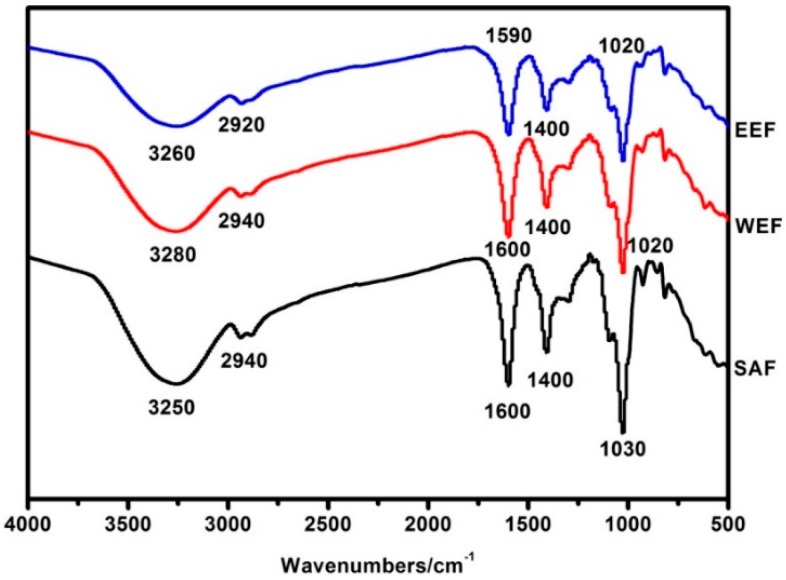
FT-IR spectra of SAF (Sodium alginate film), WEF (Sodium alginate incorporating water extract film) and EEF (Sodium alginate incorporating ethanolic extract film).

**Figure 3 materials-12-02923-f003:**
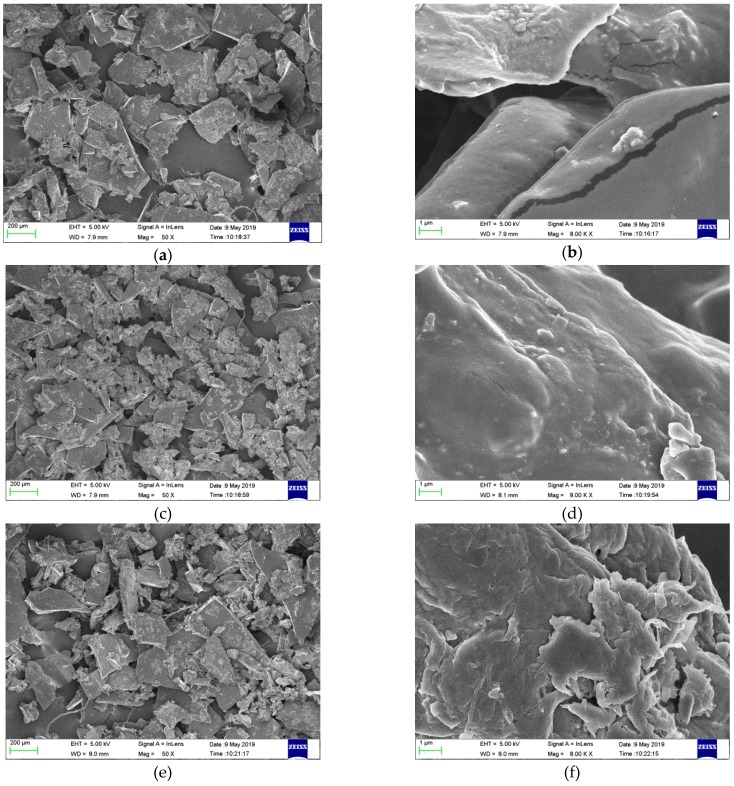
SEM micrographs of films: **a**, **c**, **e**, **g**, **i**, **k**, **m** (SAF, 10% WEF, 15% WEF, 20% WEF, 10% EEF, 15% EEF and 20% EEF, magnification: 50×) and **b**, **d**, **f**, **h**, **j**, **l**, **n** (SAF, 10% WEF, 15% WEF, 20% WEF, 10% EEF, 15% EEF and 20% EEF, magnification: 8000×).

**Figure 4 materials-12-02923-f004:**
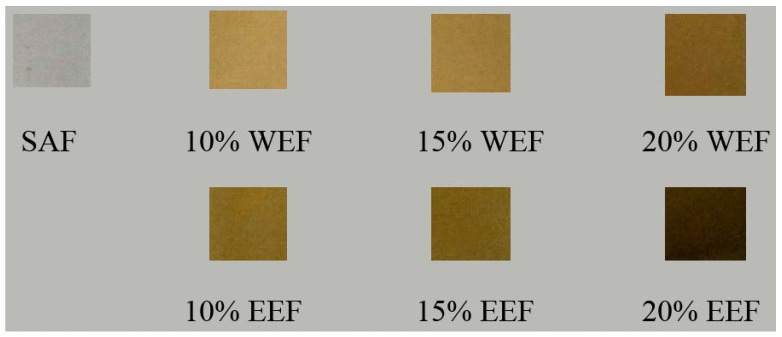
Physical appearances of the different films.

**Figure 5 materials-12-02923-f005:**
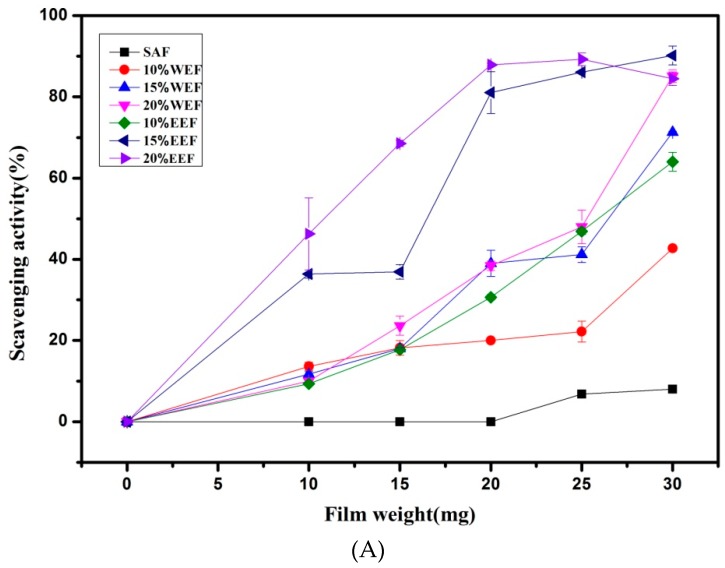
Weight-dependent (**A**) and time-dependent (**B**) DPPH radical scavenging ability of the films.

**Figure 6 materials-12-02923-f006:**
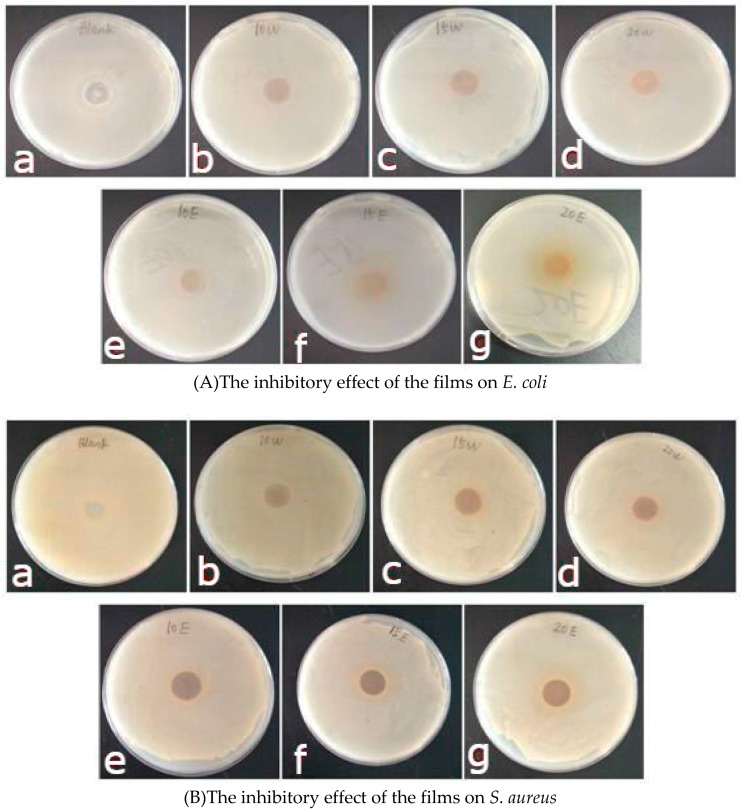
The antibacterial activity of the different films against *E. coli* (**A**) and *S. aureus* (**B**). (**a**: SAF, **b**: 10%WEF, **c**: 15%WEF, **d**: 20%WEF, **e**: 10%EEF, **f**: 15%EEF, **g**: 20%EEF).

**Table 1 materials-12-02923-t001:** The formulations of the films.

Sample	1%SA + 0.5%Gly (mL)	WE (mL)	EE (mL)
SAF	100	0	0
10%WEF	90	10	0
15%WEF	85	15	0
20%WEF	80	20	0
10%EEF	90	0	10
15%EEF	85	0	15
20%WEF	80	0	20

**Table 2 materials-12-02923-t002:** Color parameters of the different films.

Films	L	A	B	ΔE
SAF	85.84 ± 0.38a	0.55 ± 0.46c	−2.57 ± 0.14e	0.49 ± 0.43e
10% WEF	75.06 ± 2.48b	2.29 ± 0.92b	24.41 ± 3.99d	29.50 ± 4.67d
15% WEF	72.81 ± 1.34b	2.87 ± 0.33b	29.26 ± 1.12c	34.86 ± 1.48c
20% WEF	69.99 ± 1.05c	4.74 ± 0.56a	34.74 ± 1.53b	41.14 ± 1.87b
10% EEF	68.12 ± 2.10cd	0.12 ± 0.74cd	34.20 ± 1.77b	41.24 ± 2.49b
15% EEF	66.10 ± 0.82d	2.37 ± 0.08b	37.01 ± 0.80ab	44.25 ± 1.06b
20% EEF	62.06 ± 0.32e	−0.39 ± 0.38d	39.62 ± 0.22a	48.88 ± 0.06a

Values are given as the mean ± SD (n = 3). Letters (a–e) indicated significant differences (*p* < 0.05).

**Table 3 materials-12-02923-t003:** Moisture content, water solubility, thickness and water vapor permeability of the films.

Films	MC/%	Solubility/%	Thickness/mm	WVP/×10^−11^ g·Pa^−1^·s^−1^ cm^−1^
SAF	66.92 ± 1.58a	76.31 ± 0.51f	0.161 ± 0.030a	1.79 ± 0.02a
10% WEF	28.97 ± 0.37d	83.49 ± 0.48e	0.087 ± 0.015b	0.95 ± 0.01b
15% WEF	36.26 ± 1.03c	89.39 ± 0.78d	0.075 ± 0.008bc	0.83 ± 0.01c
20% WEF	38.49 ± 0.50b	98.46 ± 0.83a	0.057 ± 0.006cd	0.63 ± 0.01e
10% EEF	20.71 ± 0.73e	92.74 ± 0.77c	0.050 ± 0.010d	0.55 ± 0.03f
15% EEF	18.18 ± 0.44f	93.89 ± 0.20b	0.055 ± 0.007cd	0.61 ± 0.02e
20% EEF	17.24 ± 0.33f	94.03 ± 0.28b	0.068 ± 0.015c	0.75 ± 0.02d

Data are presented as the mean ± SD (n = 6 for thickness, n = 3 for MC, solubility and WVP). Letters (a–f) indicated significant differences (*p* < 0.05).

**Table 4 materials-12-02923-t004:** Mechanical properties of the films.

Films	Tensile Strength/MPa	Elongation/%
SAF	1.11 ± 0.16d	88.16 ± 1.29a
10% WEF	10.86 ± 0.36b	22.47 ± 2.39b
15% WEF	9.39 ± 0.06b	17.82 ± 1.24c
20% WEF	5.90 ± 0.29c	6.78 ± 0.83d
10% EEF	17.15 ± 1.19a	6.38 ± 1.05d
15% EEF	17.36 ± 1.72a	5.44 ± 1.97d
20% EEF	8.52 ± 1.53b	5.36 ± 1.62d

Values are given as the mean ± SD (n = 3). Different letters (a–d) indicated significant differences (*p* < 0.05).

**Table 5 materials-12-02923-t005:** Antimicrobial properties of the films.

Films	Inhibition Zone Diameter (mm)	
	*E. coli* (−)	*S. aureus* (+)
SAF	13.83 ± 0.16d	14.00 ± 0.46d
10% WEF	14.08 ± 0.25d	15.85 ± 0.38c
15% WEF	14.52 ± 0.23c	16.30 ± 0.42c
20% WEF	15.17 ± 0.26c	16.33 ± 0.39c
10% EEF	15.47 ± 0.45c	16.03 ± 0.37c
15% EEF	21.17 ± 0.35b	17.22 ± 0.40b
20% EEF	23.20 ± 0.36a	18.22 ± 0.30a

Values are given as the mean ± SD (n = 3). Different letters (a-f) indicated significant differences (*p* < 0.05).

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
