# Peer review of "Sodium Alginate-Based Green Packaging Films Functionalized by Guava Leaf Extracts and Their Bioactivities"

_materials, 2019, doi:10.3390/ma12182923_

Round 1
Reviewer 1 Report
The authors have studied the possibility of formation of sodium alginate-based green packaging films functionalized by guava leaf extracts and their bioactivities.
In my opinion, the work has been structured in a linear way with good clarity. However, the antibacterial testing should be described more precisely. Moreover, results could be supported with more references. The authors should explain/describe the following points:
Line 64- characterize sodium alginate, catalogue number, viscosity, molecular weight etc. Glycerol and ethanol are missing
Line 72- How guava leafs were stored before use?
Line 85- what was the reason of so high temperature? What was the speed of stirring and type of used equipment?
Line 88- authors should add a table with the film formulations
Line 95- for how long films have been stored?
Line 177- there is no As in the equation
Line 180- write this part more precisely
Line 184- how authors measured 106 CFU/mL of microorganisms?
Line 186- how the inhibition zones were measured? Did you observe the growth below the film disk?
Line 195- authors should summarize obtained results.
Fig 2- the quality of the figure 2 needs to be improved
Line 274- Can color of films be a limitation of their use?
Line 361- Did the authors performed the analysis after a longer time of storage? Is 60 minutes enough to make a statement as in line 377?
Line 398- Change the title of the table to Antimicrobial properties of films.
Line 386- Add the figures of Petri dishes where the inhibition zone can be observed.
Line 403- Are you sure that WE greatly enhanced the antibacterial activity of SA films?
Line 406- Please explain more precisely why 10% EEF sample is the best in your opinion? I do not agree with this statement.
Author Response
In my opinion, the work has been structured in a linear way with good clarity. However, the antibacterial testing should be described more precisely. Moreover, results could be supported with more references. The authors should explain/ describe the following points:
Response: Thank you very much for your valuable suggestions. We have made modifications according to your suggestion in the revised paper.
Line 64- characterize sodium alginate, catalogue number, viscosity, molecular weight etc. Glycerol and ethanol are missing
Response: Thanks a lot for your suggestion. We are sorry for our carelessness. Details of these reagents have been added in the revised manuscript.
Line 72- How guava leafs were stored before use?
Response: Thanks a lot for your suggestion. Guava leaves were naturally dried and stored in dark at room temperature (25±2 °C).
Line 85- what was the reason of so high temperature? What was the speed of stirring and type of used equipment?
Response: Thanks a lot for your suggestion. We used a B11-2 constant temperature magnetic stirrer (Shanghai Sile Instrument Co., Ltd.). The temperature was set at 90℃, the speed of stirring was 450r/min. The actual temperature of the solution was 60±1 ℃. The selection of temperture referred to the previously repoeted work by Yang et al.
Line 88- authors should add a table with the film formulations
Response: Thanks a lot for your suggestion. We have added a table with the film formulations.
Line 95- for how long films have been stored?
Response: Thanks a lot for your suggestion. Films were stored at 20 °C and 50% RH in a constant temperature and humidity incubator for 3 days before further tests.
Line 177- there is no As in the equation Response:
Thanks a lot for your suggestion. We have revised the equation.
Line 180- write this part more precisely
Response: Thanks a lot for your suggestion. We have made some modifications in this part.
Line 184- how authors measured 106 CFU/mL of microorganisms?
Response: The number of microorganisms was calculated by hemocytometer.
Line 186- how the inhibition zones were measured? Did you observe the growth below the film disk?
Response: We observed the growth below the film disk and there was no bacteria. The diameter of inhibition zones including the film disk was measured with a caliper and recorded in millimeters.
Line 195- authors should summarize obtained results.
Response: Thanks a lot for your suggestion. We have summarized the results of this part.
Fig 2- the quality of the figure 2 needs to be improved
Response: Thanks a lot for your suggestion. The quality of the figure 2 was improved in the revised manuscript.
Line 274- Can color of films be a limitation of their use?
Response: Thanks a lot for your question. Yes, the color of films can limit their use. These colored films might be suitable for inner packaging or used for foods susceptible to light exposure. We can choose appropriate packaging materials according to the product packaging requirements.
Line 361- Did the authors performed the analysis after a longer time of storage? Is 60 minutes enough to make a statement as in line 377? Response: We did not performed the analysis after a longer time of storage. The scavenging rate of the films reached the maximum at 40 min or 50 min. Thanks a lot for your suggestion. More comprehensive monitoring the antioxidant activity of the films such as storage time and free radical concentration in our further study.
Line 398- Change the title of the table to Antimicrobial properties of films. Response: Thanks a lot for your suggestion. We are sorry for our carelessness. We have changed the title of Table 5 to Antimicrobial properties of films.
Line 386- Add the figures of Petri dishes where the inhibition zone can be observed.
Response: Thanks a lot for your suggestion. We have added the figures of antimicrobial tests in revised manuscript.
Line 403- Are you sure that WE greatly enhanced the antibacterial activity of SA films?
Response: Thanks a lot for your suggestion. We carefully checked the data and the results of variance analysis. The results indicated that 10% WE significantly enhanced the inhibitory effect of SA film on S. aureus (p<0.05), and 15% WE significantly enhanced the inhibitory effect of SA film on E.coli (p<0.05). We have revised the expression of this part.
Line 406- Please explain more precisely why 10% EEF sample is the best in your opinion? I do not agree with this statement.
Response: Thanks a lot for your suggestion. We revised the conclusion after a comprehensive comparison.

Reviewer 2 Report
The manuscript can be published in Materials after a major revision. The comments are as below:
The introduction section should be improved, there are lots of researches regarding utilization of alginate-based films and coatings for food packaging applications, the authors should be reviewed these papers and explain the advantage and novelty of the current work. In the result section, the results should be compared with the reported values in the literature to show the advantage of current alginate-based films. As can be seen, functionalization by guava leaf extracts can not significantly improve the properties of alginate films. What is the effect of the initial concentration of alginate solution on the properties of the films? Increasing the initial concentration how can affect the final properties of the films? What is the effect of glycerol in the preparation of the films? Addition of glycerol can improve the stability or mechanical properties of the film? There are so many papers which reported preparation of alginate-based film without the addition of plasticizer, what is the advantage of using plasticizer? Crosslinking of the films can improve the mechanical and other properties of the films, why the authors do not try this?Author Response
The manuscript can be published in Materials after a major revision. The comments are as below:
The introduction section should be improved, there are lots of researches regarding utilization of alginate-based films and coatings for food packaging applications, the authors should be reviewed these papers and explain the advantage and novelty of the current work.
Response: Thanks very much for your valuable suggestion. According to your suggestion, we have reviewed the related papers and made some modifications in this part.
In the result section, the results should be compared with the reported values in the literature to show the advantage of current alginate-based films.
Response: Thanks very much for your valuable suggestion. We have added comparative discussion in revised manuscript.
As can be seen, functionalization by guava leaf extracts can not significantly improve the properties of alginate films. What is the effect of the initial concentration of alginate solution on the properties of the films? Increasing the initial concentration how can affect the final properties of the films?
Response: Thanks very much for your valuable suggestion. The tensile strength and elongation at break of the film increased with the increase of sodium alginate concentration in the film-forming solution. The film thickness also increased gradually. The concentration of sodium alginate was too high, the elongation at break decreased and. Our preliminary experiments showed that the suitable concentration of sodium alginate was between 0.8% and 1.2%.
What is the effect of glycerol in the preparation of the films? Addition of glycerol can improve the stability or mechanical properties of the film? There are so many papers which reported preparation of alginate-based film without the addition of plasticizer, what is the advantage of using plasticizer? Crosslinking of the films can improve the mechanical and other properties of the films, why the authors do not try this?
Response: Thanks very much for your valuable suggestion. Addition of glycerol can improve the mechanical properties and water resistance of the film. I agree with you that crosslinking of the films can also improve the mechanical and other properties of the films. This work is being considered and will be carried out in the future.

Reviewer 3 Report
The manuscript entitled 'Sodium alginate-based green packaging films functionalized by guava leaf extracts' described different formulations for development of bioactive films with sodium alginate and guava leaf extracts. These biofilms have antioxidant and antibacterial activity. The manuscript is well written but need to address few suggestions to improve the quality before publication.
Comments:
In introduction section, describe why you have selected guava leaf? Provide the source and botanical name guava that you have used in this study. Discuss the previous work and discuss the gap between previous and your work. Include the references for techniques used for analysis. Start a sentence with 'The' instead of number.Line 182: correct Escherichia Coli, first letter of species.
Line 199: equivalent/1 mL extract, correct it. Line 338:Different letters (a-f) indicated significant differences (p<0.05). Please write the meaning of a-f. A quality of figure 5 can be improved.Author Response
The manuscript entitled 'Sodium alginate-based green packaging films functionalized by guava leaf extracts' described different formulations for development of bioactive films with sodium alginate and guava leaf extracts. These biofilms have antioxidant and antibacterial activity. The manuscript is well written but need to address few suggestions to improve the quality before publication.
Comments:
In introduction section, describe why you have selected guava leaf? Provide the source and botanical name guava that you have used in this study. Discuss the previous work and discuss the gap between previous and your work. Include the references for techniques used for analysis. Start a sentence with 'The' instead of number.
Response: Thanks a lot for your suggestion. We have made modifications followed your suggestions in the revised manuscript.
Line 182: correct Escherichia Coli, first letter of species.
Response: Thanks a lot for your suggestion. It was corrected.
Line 199: equivalent/1 mL extract, correct it.
Response: Thanks a lot for your suggestion. It was corrected.
Line 338:Different letters (a-f) indicated significant differences (p<05). Please write the meaning of a-f.
Response: Thanks a lot for your suggestion. We added the meaning of a-f in 2.5.
A quality of figure 5 can be improved.
Response: Thanks a lot for your suggestion. The quality of figure 5 was improved.

Round 2
Reviewer 1 Report
Manuscript was improved. I recommend it for publication.
Reviewer 2 Report
The manuscript can be published in this form.